# Research on the Evaluation Index System of the Construction of Communities Suitable for Aging by the Fuzzy Delphi Method

**Wen-Bing Mei [1], Che-Yu Hsu [2] and Sheng-Jung Ou [2],***

[1]  Department of Art Design, Guangdong Industry Polytechnic, Guangdong 528225, China; mlkun3101@gmail.com
[2]  Department of Landscape and Urban Design, Chaoyang University of Technology, Taichung 413, Taiwan; cyhsu428@cyut.edu.tw
*   Correspondence: sjou@cyut.edu.tw

**Abstract:** In order to cope with the rapidly aging society and the Chinese traditional idea of old-age care, community home-based care has become a major mode of care for the aged in China, and the construction of communities suitable for the aged has also become the focus of the whole society. In order to build an objective and scientific evaluation index system of communities suitable for aging, the hierarchical structure and relative important values of the indicators for the assessment were obtained through data text rooted coding and the fuzzy Delphi expert questionnaire survey. The results prove that: the evaluation index system of communities suitable for aging consists of 4 criteria (i.e., humanistic care, public environment, health care, and economic security), 14 sub-criteria, and 48 evaluation indexes. The important values of the four criteria are "health care" (7.35), "public environment" (7.18), "humanistic care" (6.92), and "economic security" (6.87). This shows that with the aging of the elderly and the decline of physical function, health care is the most important material basis for community home care, and is also the core criterion for the construction of communities suitable for aging. Of the 48 important values of the evaluation indexes, "setting up an emergency assistance system" (7.89), "ensuring the accessibility of roads" (7.79), and "handling related affairs" (7.60) ranked the highest. This shows that the protection of the physical and mental health of the elderly and the rights and interests of the elderly are the core values of the construction of communities suitable for aging. This study believes that through constructing an evaluation index system of the communities suitable for aging, the past general constructions suitable for aging according to the standard can be further clarified through the scientific procedure of the 'community suitable for aging construction evaluation method', and be a reference for the academic and practical fields.

**Keywords:** community; home-based care; aging; evaluation index; important values; fuzzy Delphi method

---

## 1. Introduction

### 1.1. Research Background and Motivation

Since China became defined as an aging society in 2000, China's population structure has been aging rapidly and getting older. "Aging" has changed from a social phenomenon to a social problem that cannot be ignored, and has put forward severe requirements for the development of China's old-age care industry. With the influence of many factors such as population policy, the urbanization process, and increasingly frequent social mobility, the family care function is gradually weakened, and the elderly are increasingly dependent on social care services. As the best combination of family and

society, the community home-based old-age care model can not only enable the elderly to provide for the aged in a familiar environment, but also enable the home-based elderly to receive social services based on the community. Therefore, it has developed into an important mode for the majority of the elderly to meet their needs for old-age care services, which plays an important role in the current old-age care service system (Tong, 2015) [1]. As an important guarantee for elderly life, the ultimate goal of community elder-care services and age-appropriate construction is to improve the quality of life and life satisfaction of the elderly. Whether the connotation of community elder-care service supply and age-appropriate construction can match with the actual needs of the elderly, and whether the diversified needs of the elderly can be effectively satisfied, are the key links that affect the development of community home-based elder-care services and age-appropriate constructions (Tu, 2016) [2]. Different from the aging process in developed countries for decades or even hundreds of years, China's rapid aging still coexists with the social background of "getting old before getting rich", which determines that the elderly's demand for care services cannot be fully met under the constraints of social endowment resources. Therefore, how to identify the needs of the elderly and adopt the corresponding strategies to build communities suitable for aging has become a problem that the current model of community home-care is faced with and urgently needs to solve (Wu and Miao, 2004) [3].

The accurate screening and priority satisfaction of the needs of the elderly depend on the judgment of the importance of different elder-care services, while the classification of the needs of the elderly can reflect differences in the importance of different elder-care services. The weakening of the family care function leads to the dependence of the elderly on the community service supply, and the actual degree of dependence determines the key factors needed for the elderly to obtain the care service behavior. Therefore, how to identify and grasp the "importance" of the elderly to the elder-care services is a strategic arrangement concerning the construction of an aging community (Wang and Liu, 2012) [4]. Therefore, focusing on the goal of meeting the needs of the majority of the elderly, as well as the important degree of identifying the needs of the elderly and by what standards is an important part of community home-based elder-care services, as well as the foothold and focus of the construction of communities suitable for aging (Liu, 2012) [5].

*1.2. Research Purpose*

To collect and code the maximum extent of the indicators for the construction of communities suitable for the aged under the model of community home-based care, through the sorting out of laws and regulations for the aged, the discussion of research literature and the rooted interview of the aged. The specific purposes of this study are as follows:

- To use the fuzzy Delphi expert questionnaire survey method to effectively screen the assessment indicators for the construction of communities suitable for aging, and construct a complete evaluation index system for the construction of communities suitable for aging;
- To discuss the ranking of the importance of the assessment indicators for the construction of communities suitable for aging with the help of the calculation of the consensus value of the expert group score;
- To provide a reference for the construction of communities suitable for aging in the field of old-age care.

## 2. Literature review

*2.1. Research on the Types and Characteristics of Elder-Care Needs*

In January 2002, *Active Aging: A Policy Framework* was published by the World Health Organization (WHO) Center for Healthy Development, which proposed the three dimensions of "health", "participation" and "security" of elder-care needs. In 2015, in the global report on aging and health, it was proposed that, on the basis of ensuring the living and health care of the elderly, attention

should also be paid to the socio-economic and human rights development of the elderly (World Health Organization, 2015) [6]. As a programmatic document in the field of global policy on the elderly, the Vienna international plan of action on aging defines the provision of housing security, medical care, social services, education and training, spiritual comfort, and other services for the elderly (Chen, 2007) [7]. Rose and Forder outlined the needs of the elderly in four areas: housing, health, economics, and psychology. Among them, housing involves the realization of the needs of old age in the living environment. Health needs involve health maintenance, chronic disease management, and daily maintenance of medical care. Economic needs involve economic assistance and subsidy measures to maintain the normal standard of living of the elderly. Psychological needs include spiritual comfort, goal achievement, and humanistic care measures (Rose and Forder, 1975) [8]. Marjorie and Little divided elder-care services into health care, auxiliary medical care, household service, transportation, spiritual care, personal affairs, and other contents from the four dimensions of material life assistance, health care assistance, social economic assistance, and personal social development assistance (Marjorie Cantor, 1991) [9]. Domestic scholars generally divide the needs of the elderly into four categories: life, health, economy, and spirit. For example, Zhou, Yan, and Zhao divided the elderly's needs for community old-age care into four dimensions: material life, daily care, health care, and spiritual culture, and believed that the elderly have the most urgent needs for medical assistance and spiritual care (Zhou, Yan, and Zhao, 2001) [10]. Wang divided the needs of the elderly into "basic needs" and "potential needs". "Basic needs" include endowment insurance, medical care, daily care, and legal rights protection. "Potential needs" include mental health, spiritual care, elderly culture, and self-realization (Wang, 2006) [11]. According to Maslow's hierarchy of needs theory, Hu divided the needs of the elderly into material old-age support and economic support to meet their physiological needs and security needs. Hu Ai Ming also identified spiritual care, so that the elderly feel affection, and neighborhood care, so that they feel respected. Cultural endowment enables the elderly to live a rich life in their old age and realize their own value and meaning of life (Hu, 2012) [12]. The elder-care needs assessment scale compiled by Wang and Bai assessed 27 items in 5 dimensions, including self-care ability, cognitive ability, emotional behavior, social function, and health maintenance (Wang and Bai, 2020) [13].

*2.2. Research on the Influencing Factors and Satisfaction of the Elderly's Care Demand*

The academic circles mainly focus on the qualitative and quantitative aspects of the influencing factors of care demand. In the qualitative analysis, Song adopted the research idea of "master position" and concluded through the rooted interview that if the elderly have a strong initiative and rational choice in their demand for care, they will choose the path or strategy to meet their demand for a pension according to their own situation, external environment, and the type and quantity of resources owned by the family (Song, 2010) [14]. Lu discussed the substitution effect of social services on family care from the perspective of "family spillover" and calculated the inverse functional relationship between family care and social services using the effect model (Lu, 2017) [15]. According to the effect model principle of classical economics, Font and Courbage took home care as the exogenous variable of the model. Through qualitative comparison and analysis, they investigated the crowding-out effect between home care and social care service demand, and verified the correctness of the conclusion derived from the model with some data from Europe (Font and Courbage, 2015) [16]. In terms of quantitative analysis, Wang pointed out that the community home-based care service generally has a situation where the supply exceeds the demand, through the urban elderly data from the 2010 *Tracking Survey of China's Urban and Rural Elderly Population* by the China Scientific Research Center On Aging at the same time. The needs of the elderly are also affected by factors such as the willingness to participate in social activities, gender, region and so on (Wang, 2010) [17]. Guo and Hao based on the data of the same group of elderly people in the tracking survey of influencing factors of the health of the elderly in China from 2005 to 2011, and using the principal component analysis method, concluded that the order of influencing factors of the elderly people's old-age care needs was different in different

periods (Guo and Hao, 2019) [18]. Using structural equation model analysis, and empirical analysis on the endowment service supply level of inner texture structure and logic, Zheng and Lee pointed out that the main influence factors of endowment demand are willingness, family characteristics, system characteristics, and the age of the subject. By using the methods of "descriptive statistics of variables" and "logit model estimation", the study concluded that care demand was positively correlated with physical function (dependence on daily activities, age) and socioeconomic status (annual income, real estate, education level). It was inversely related to cultural factors (thrift, consideration for children) and family factors (number of children, living style). It is an important measure to pay attention to the public supply of old-age services and improve the level of building communities suitable for aging, which can effectively improve the well-being of the elderly (Zheng and Lee, 2017) [19]. The WHO project proposed that an age-friendly city is one that promotes active aging and optimizes opportunities for health, participation, and security, in order to enhance quality of life as people age. The features of age-friendly cities were determined in eight domains of urban life, namely outdoor spaces and buildings; transportation; housing; social participation; respect and social inclusion; civic participation and employment; communication and information; and community support and health services (J. van Hoof and J. K. Kazak, 2018) [20].

### 2.3. Research on Aging Related Indicators and Evaluation Methods

The concept of healthy and active aging was defined by the World Health Organization (WHO) as the process of optimizing opportunities for health to enhance quality of life as people age. The word "healthy" refers to physical, mental, and social well-being, while the word "active" refers to continuing participation in social, economic, cultural, spiritual, and civic affairs (Bousquet, Kuh, Bewick, Standberg, Farrell and Pengelly, 2015) [21]. To design policies to promote healthy and active aging, as well as to track their progress, measurement is crucial. With that in mind, the European Commission (EC) and the United Nations Economic Commission for Europe introduced the Active Aging Index (AAI) in 2012. The AAI is a multidimensional index that measures the level to which older people (55+) live independent lives and participate in the labor market and social activities, as well as their capacity to actively age (UNECE, 2014) [22]. The AAI provides a societal perspective of the aging phenomenon and is a useful tool for top-down policy design. However, it only allows for the comparison of average levels of active aging across countries. Designing policies focused on the most vulnerable groups requires information about the distributions of healthy and active aging within countries, as well as information about how healthy and active aging correlates with individual characteristics. To obtain such information, we need to measure healthy and active aging at the individual level (Barros and Almeida, 2015) [23]. The active aging index is used to measure and compare countries' progress and levels in four aspects: employment, social participation, healthy, independent and safe living, and active aging environment and capacity (Barslund, Werder and Zaidi) [24]. The active aging index, as a composite index which can effectively quantify the development level of active aging, has attracted extensive attention in academic circles. Liu Wen and Yang Fuping used the CHARLS and CGSS database, drew on the EU active aging measurement framework, and designed the active aging index of China by combining AHP and DEA. It measures the active aging index of China's three regions and 28 provinces, and studies the active aging development level at regional, urban and rural, provincial and gender levels. The results show that the development of active aging in China is unbalanced, showing a trend of high–east and low–west. The difference of active aging level between urban and rural areas is obvious, the male level is generally higher than that of female, and the gap is gradually widening with the growth of age (Liu and Yang, 2019) [25].

In view of the difficulties in examining the health and active aging of individuals, Judite and Maria offer an index of spontaneous aging (SAI), that is, policies aimed at the most vulnerable groups require information at the individual level (Judite and Maria, 2017) [26]. SAI is based on a conceptual framework that attempts to capture the health and active aging of individuals. Its basic approach allows weights to be determined by variations and correlations in the data, thereby avoiding value

judgments (Sirven and Debrand, 2014) [27]. SAI is also innovative in that it can be fully self-evaluated, allowing us to consider SAI as a future tool for older people to track their aging status. SAI is based on the Biological–Psychosocial Assessment Model (MAB), a tool used to make multidimensional assessments of older persons in three dimensions: biological, psychological and social. The indicators were selected and the weights were determined according to an ordered probability model, which correlated the MAB indicators with self-assessment indicators to self-assess health and active aging (Steptoe, Wright, Kunz-Ebrecht and Iliffe, 2006) [28].

*2.4. Comprehensive Evaluation of Literature Research*

To sum up, there is no unified standard for the classification of the elderly's care needs in the academic community. The types of care needs are not only the same based on different perspectives, which also fully demonstrates the complexity and diversification of the elderly's care needs. At the same time, the study found that no matter what standard classification was adopted, the elderly's needs for old-age care all covered the living environment, health care, financial support, and spiritual care in the daily care. Therefore, in order to ensure the integrity of community home-based care services and the rationality of the construction of community suitable for aging, this paper will analyze the elderly needs of community home-based care according to the classification of humanistic care, public environment, health care, and economic security.

Similarly, for a long time, domestic and foreign scholars have conducted a rich survey and research on the influencing factors and satisfaction of care demand. In general, the existing studies on the influencing factors of the elderly's pension needs and their satisfaction are limited to the elderly's physical, psychological, and economic resources, while little attention has been paid to the construction of communities suitable for aging. Therefore, the existing research conclusions do not play a strong role in supporting the improvement of community services and the construction of communities suitable for aging. The construction of communities suitable for aging is closely related to the life of the elderly, which is the core of the community home-care service system. Although some scholars have put forward the research on the influencing factors and satisfaction of elder-care needs from the perspective of a livable living environment for the elderly and the use of Internet technology, the research focuses on different aspects. However, no in-depth analysis has been conducted on how the elderly community home-care satisfaction is affected by the aging environment in the community. In view of this, this paper intends to start from the elder-care needs of community home care, through the combination of in-depth interviews and fuzzy Delphi expert questionnaire survey, to analyze the demand characteristics of community home care services and the methods of resource allocation, in order to provide a certain scientific support for the construction of communities suitable for aging.

**3. Materials and Methods**

Based on qualitative research and quantitative analysis, this paper probes into the indicators of the construction of communities suitable for aging under the model of community home care. Research was done in two stages. The first stage collected and coded the community home endowment mode through text analysis of the construction of the optimal aging indicators. The second stage used the fuzzy Delphi expert questionnaire to screen the community aging of the construction of the evaluation index, so as to construct a complete community optimal aging evaluation index system. The study was conducted between September 2019 and May 2020.

*3.1. The First Stage: Collection and Coding of Indicators for the Construction of Communities Suitable for Aging under the Model of Community Home Care*

In order to ensure the objectivity and relative integrity of the construction index of community fitness for aging under the current model of community home-based care, and to reduce the deviation caused by subjective judgment, this stage adopts qualitative research to clarify the index framework of community fitness for aging based on community home-based care. Under the guidance of exploratory

research, the analysis of the content of community aging construction under the community home-based care model is the core content related to the elder-care needs of the community, and is also the first step of this study. The analysis material comes from three aspects—policy text, research literature, and interview material, so as to cover the possible elder-care needs under the model of community home care to the greatest extent. The application of the literature review method to evaluate and classify the collected data can improve the understanding of the wide range of the research topic, and give a glimpse of the research trend and future exploration fields (Wang, 2003) [29]. From the subjective point of view of the elderly, through interviews with the elderly and relevant practitioners, we can understand the elderly's needs and the defects in the construction of their communities, which can effectively alleviate the current lack of research on the needs of the elderly from their own perspective (Tian and Meng, 2012) [30]. Based on Strauss's grounded theory, this paper divides the data analysis process into three steps: open, axial, and selective coding. The encoding generation of qualitative analysis generally has two ways: induction and deduction. However, considering the particularity of this paper, the "synthesis method" proposed by Miles and Huberman is adopted for coding; that is, a basic explanation is proposed in advance based on the combination of deduction and induction, and then the code is established in this system (Miles and Huberman, 1994) [31]. This study will use national policy text coding, coding of documents and materials, and interview materials coding after merger, combining these with literature reviews of previous studies. The community endowment of aging that occupies the home demand is divided into four standard layers, using the "comprehensive" coding procedures, through the freedoms node class relations, in the form of the construction of the community home endowment mode suitable aging index framework. The coding process is completed with the assistance of NVivo12.0. Through the bottom-up induction process, the nodes at all levels from the bottom to the top of the dependency relationship are finally formed. NVivo was developed by an Australian QSR company, which has the functional orientation of grounded theory constructivism. It is applicable to qualitative research with the combination of behavior, content, discourse analysis, and longitudinal research, literature review and multiple methods. It is widely used in information science, psychology, education and other fields (Hu, 2012) [32].

### 3.1.1. Policy Text Analysis

In order to ensure that the community is suitable for aging construction, this paper first sorted out the community home-care services involved in the policy. Through the legal database of Peking University (pkulaw.cn), taking "community home care for the aged; community construction; elderly care service" as the keywords, the national policies were selected one by one in an "accurate and vague" way. A total of 149 policy texts from 2000 to 2019 were selected. The policy themes cover the fields of economic security, culture and education, medical and health care, environmental construction, and financial maintenance. Based on the semantic analysis and keyword query of the above policies, the coding results obtained a total of 43 free nodes (Table 1). These nodes are located at the bottom of the subordinate relationship and are the direct factors that affect the construction of communities suitable for aging.

**Table 1.** Free node information for the policy text.

| Node Name | Source | Reference Point | Node Name | Source | Reference Point |
|---|---|---|---|---|---|
| Convenience service outlets | 61 | 232 | Preferential rides for the elderly | 11 | 29 |
| Tag system | 40 | 141 | Nursing home for the elderly | 17 | 25 |
| Legal policy publicity | 11 | 140 | Rehabilitation home for the elderly | 14 | 24 |

**Table 1.** *Cont.*

| Node Name | Source | Reference Point | Node Name | Source | Reference Point |
|---|---|---|---|---|---|
| Improving the driving environment | 58 | 125 | Senior professional access | 18 | 23 |
| Public transit system | 20 | 124 | Daily living environment | 9 | 23 |
| Traffic signs | 10 | 124 | Day care center | 10 | 20 |
| Accessibility of public facilities | 22 | 65 | Sociocultural environment | 8 | 20 |
| Internet retirement | 12 | 64 | Community participation | 15 | 20 |
| Family doctor | 17 | 51 | Community walk | 12 | 18 |
| Health support | 18 | 48 | Community security | 6 | 17 |
| Health guidance assessment | 15 | 42 | Daily care | 1 | 17 |
| Traffic safety facilities | 27 | 41 | Cultural and sports facilities | 7 | 16 |
| Emergency rescue system | 10 | 40 | Environmental accessibility | 8 | 15 |
| Spiritual consolation | 19 | 34 | Fire safety | 12 | 14 |
| Living condition | 21 | 32 | Communication for information | 10 | 13 |
| Elderly service facilities | 15 | 42 | Leisure facilities | 8 | 13 |
| College for the aged | 27 | 42 | Medical service outlets | 9 | 12 |
| Aging education | 10 | 40 | Optimize the medical environment | 10 | 12 |
| Senior activity center | 19 | 34 | Friendly transportation | 9 | 11 |
| Meals for the elderly | 21 | 32 | Volunteer service | 8 | 11 |
| Goods management for the elderly | 15 | 32 | Giving medicine dispensing | 8 | 10 |
| Senior residence | 17 | 32 | | | |

### 3.1.2. Literature Analysis

This research uses the Internet as the search tool, and searches the core journals and CSSCI based on the CNKI database. In order to ensure the integrity of literature collection, "elderly community", "community home care", "old community environment", "construction of livable communities for the elderly", and "living environment of aging communities" are the keywords for cross-screening. A total of 78 literatures related to community home-care needs and community construction were selected. The total number of free nodes is 90. The results are shown in Table 2.

**Table 2.** Free node information of literature materials.

| Node Name | Source | Reference Point | Node Name | Source | Reference Point |
|---|---|---|---|---|---|
| Mark clearly | 13 | 20 | Day care services | 15 | 47 |
| Community participation | 16 | 226 | Pedestrian access management | 6 | 9 |
| Orderly vehicle management | 1 | 3 | Day care center | 16 | 47 |
| Road lighting | 6 | 8 | Housekeeping service | 3 | 8 |
| The ground is flat and slippery | 13 | 23 | Family doctor | 2 | 4 |
| Regular body check | 3 | 4 | Well-equipped facilities | 10 | 38 |
| Short-term custody | 2 | 3 | Social participation | 20 | 54 |
| Rights protection by law | 16 | 53 | Social welfare | 17 | 43 |
| Control the step space reasonably | 29 | 447 | Social interaction | 13 | 34 |
| Internet person | 4 | 28 | Community walking environment | 7 | 13 |
| Safety in outdoor activities | 19 | 115 | Community road environment | 6 | 8 |
| Outdoor activity space | 8 | 34 | Community service identity | 2 | 10 |
| Nursing care | 40 | 257 | Community service construction | 28 | 114 |
| Environmental health | 4 | 6 | Community public green space | 8 | 21 |
| Environment daintiness | 9 | 12 | Community interaction | 3 | 7 |
| Activity convenient | 2 | 2 | Community space accessibility | 15 | 120 |
| Group travel | 10 | 22 | Convenient transportation | 4 | 4 |
| Health management | 3 | 6 | Community traffic orientation | 2 | 2 |
| Fitness equipment | 8 | 13 | Community greening | 4 | 5 |
| Transportation | 4 | 8 | Community autonomy | 9 | 47 |
| Emergency rescue | 14 | 28 | Community hearing meeting | 2 | 2 |
| Economic support | 49 | 362 | Community lighting | 15 | 60 |
| Spiritual consolation | 32 | 128 | Life service outlets | 6 | 9 |
| Spiritual needs | 5 | 9 | Life care | 3 | 4 |
| Comfortable living | 3 | 4 | Living facilities | 19 | 50 |
| Place of meeting and entertainment | 13 | 32 | Daily care | 21 | 88 |
| Rehabilitation care | 36 | 282 | Physical environment | 5 | 6 |
| Sustainability | 5 | 6 | Fire infrastructure | 1 | 1 |
| Space lighting without noise | 10 | 22 | Sanitary cleaning service | 41 | 191 |

**Table 2.** *Cont.*

| Node Name | Source | Reference Point | Node Name | Source | Reference Point |
|---|---|---|---|---|---|
| Spatial orientation | 13 | 19 | Cultural education | 2 | 3 |
| Easy identification of space | 10 | 11 | Entertainment | 17 | 42 |
| Space accessibility | 2 | 2 | Cultural and sports venues | 21 | 60 |
| Catering for the elderly | 12 | 16 | Cultural and sports facilities | 3 | 5 |
| College for the aged | 7 | 16 | Wheelchair accessible passage | 33 | 145 |
| Space ventilation and lighting | 13 | 37 | Mental health care | 12 | 28 |
| Facilities for senior citizens | 22 | 70 | Leisure chair | 15 | 86 |
| Senior Citizen Activity Center | 26 | 66 | Pension coupon | 5 | 5 |
| Education care | 35 | 160 | Retirement service station | 3 | 5 |
| Cultural construction | 7 | 23 | Pension network information | 4 | 12 |
| Elderly house | 24 | 79 | Elderly care information service | 5 | 10 |
| Internet equipment for the elderly | 31 | 105 | Health care | 18 | 44 |
| Habitable environment | 13 | 169 | Medical insurance | 6 | 10 |
| Chat and walk | 25 | 81 | Medical service | 7 | 20 |
| Neighborhood | 26 | 62 | Policy propaganda | 14 | 28 |
| Full life cycle design | 21 | 44 | Public security maintaining | 7 | 17 |

### 3.1.3. Interview Data Analysis

The random sampling method commonly used in empirical research is conducive to ensure a reasonable probability distribution. However, in qualitative research, the lack of elasticity and representativeness of random sampling may affect the reliability of the research results (Chen, 1996) [33]. In order to avoid the representativeness limitation of random sampling, this study adopted the intentional sampling method to interview the elderly with different ages, physical conditions, and family structure, and other representative elderly with care needs and related personnel. In the design of the interview, considering the differences in the level of community elder-care services and elder-care needs, a semi-structural interview was adopted. A brief outline was designed according to the community's needs for home-based care, such as environmental facilities, medical care, spiritual care, and economic support. The interview was controlled and guided as a whole, and the consistency of the interviewees' answers was ensured as much as possible, so as to obtain more information under the real and natural conditions. A total of 13 senior citizens were interviewed in this study. Based on the interview materials, a total of 42 free nodes were formed after similar service demands were summarized, as shown in Table 3.

**Table 3.** Free node information of interview data.

| Node Name | People Counting | Node Name | People Counting |
|---|---|---|---|
| Neighborhood | 6 | Handle affairs | 4 |
| Restaurant for the elderly | 10 | Meals for the elderly | 10 |
| Community clinic | 12 | Shopping facilities | 6 |
| Community organizing activities | 10 | Nutritious diet for the elderly | 9 |
| Outdoor activity space | 10 | Internet technology education | 8 |
| Indoor activity center | 8 | Accessibility facilities | 10 |
| Leisure chair | 8 | Road is flat and level | 9 |
| Chat and walk | 10 | Dividing area of sound | 8 |
| Son endowment | 1 | Road skid | 7 |
| House-for-pension scheme | 1 | Roadway lighting | 6 |
| Family mediation | 5 | Public green | 10 |
| Community clinic | 12 | Psychological guidance | 5 |
| Elderly monitoring system | 5 | Emotion management | 6 |
| Emergency rescue system | 6 | Rehabilitation guidance | 7 |
| Culture of filial piety | 2 | Medical equipment leasing | 3 |
| Independent space | 3 | Convenient transportation | 8 |
| Catering for the elderly | 10 | Skill training | 3 |
| Accompanying doctor | 5 | Information access | 5 |
| Internet technology | 7 | Service information bulletin | 6 |
| Multimedia equipment | 7 | Elderly volunteers | 8 |
| Policy to preach | 8 | Elderly activity centre | 10 |

### 3.1.4. Community Suitable for Aging Construction Evaluation Indicators

In this study, based on the free node setting of policy, literature, and interview materials, the expert team was invited to code the assessment indicators of community fitness for aging in the first stage. The leader of the expert team is a doctoral supervisor in design, with experience in design research, design practice, and design management. After the first phase of the expert team coding discussion, a total of 175 free nodes were collected from this research. Through the tree node function NVivo, correlations of free nodes with similar functions finally built up the 4 "humanistic care", "public environment", "health", and "economic security" levels of coding, 14 secondary coding levels, and a total of 51 free nodes of the evaluation index architecture, as the basis of the second phase expert questionnaire evaluation index selection, as shown in Table 4.

**Table 4.** Assessment indicators and interpretation of community building suitable for aging.

| Coding Content | Evaluation Index | Index Definition |
|---|---|---|
| **Humanistic care** | Community activity | |
| | Participating in community management | Encouraging the elderly to participate in the management of community residents' public affairs |
| | Organizing community activity | With the help and support of the community, the elderly can organize community activities to activate the atmosphere of community care |
| | Exchanging community information | Through community activities, the elderly can exchange and discuss information related to community service management |
| | Improving community services | On the basis of relevant policies and laws, the elderly in the community need to improve community services for the elderly |
| | Spiritual care | |
| | Providing spiritual comfort | Providing the necessary emotional communication and community communication for the elderly enrich the spiritual life of the elderly |
| | Improving neighborhood relations | Expanding community communication platforms for the elderly, coordinating neighborhood disputes, and establishing good neighborhood relations |
| | Providing psychological guidance | Providing community old people with psychological treatment and counseling services, paying attention to the old people's daily psychological health care |
| | Fostering community identity | Through the design and setting of community material space and the improvement of community functions, the community culture can be effectively constructed and the community identity of the elderly can be cultivated |
| | Cultural education | |
| | Providing cultural education | We will provide cultural re-education for the elderly in the community, meet their educational needs and improve their quality of life |
| | Providing skill training | To provide skills training for the elderly in the community, to enhance the survival and development of the elderly, to meet the needs of the elderly |
| | Organizing cultural performances | Organize community elderly people to carry out recreational and sports performances, and encourage elderly people to carry out recreational and sports activities |
| | Rights protection | |
| | Handling related affairs | Acting for the elderly in the community to handle various affairs, improve the efficiency of the elderly, to protect the daily rights and interests of the elderly |
| | Publicity policy | Through the publicity of relevant policies, to ensure that the community elderly can correctly understand the policy connotations in a timely manner, to protect the policy rights and interests of the elderly |
| | Managing supplies for the elderly | To provide guidance for the community elderly in their accessories consumption behavior, to ensure the elderly's consumption rights and interests |
| | Provide legal aid | To provide legal assistance to the elderly in the community to ensure their legal rights and interests |

**Table 4.** *Cont.*

| Coding Content | Evaluation Index | Index Definition |
| --- | --- | --- |
| **Public Environment** | Community facility | Improving fitness facilities | Improve the community fitness facilities to meet the fitness needs of the elderly |
| | | Providing leisure seating | Leisure seats are provided in the community to meet the needs of the elderly for leisure and rest |
| | | Improving lighting facilities | Improve the lighting facilities in the community to ensure the safety of the elderly's lighting needs |
| | | Improving sanitation facilities | Improve the community's public health facilities to ensure the health needs of the elderly |
| | Community road | Road accessibility | The road surface, ramps, and other barrier-free settings facilitate the normal use and safe passage of the elderly |
| | | Management of community vehicles | Orderly management of community vehicles and people, to ensure the safety of community elderly activities without interference |
| | | Improving community roads | Improve the community walking access and beautify the road environment to meet the walking needs of the elderly |
| | Built environment | Improving physical environment | The building for the elderly is well ventilated, with plenty of daylight, quiet and noiseless, meeting the needs of the elderly |
| | | Building facilities for the elderly | The steps, handrails, passageways and other barrier-free settings of the building are designed to facilitate the normal use and passage of the elderly |
| | | Space static separation | The separation of activity space and movement can meet the diversified needs of the elderly |
| | | Formulate safety measures | Ensure community safety, maintain community public order and stability, and ensure a safe living environment for the elderly in the community |
| | Outdoor environment | Beautify community environment | Community public greens are a good, beautiful environment, to meet the needs of the elderly community environment |
| | | Ensure space accessibility | The function of each area in the community is clearly marked, the community road is continuous, and the indicator system is clear to meet the accessibility needs of the elderly |
| | | Supporting living facilities | Community around the life-service network is complete, including shopping facilities, to ensure the convenience of elderly people's daily life |
| | | Ensure public transportation | Public transport facilities are perfect, and various modes of transportation are available to ensure convenient transportation for the elderly in the community |

**Table 4.** *Cont.*

| Coding Content | Evaluation Index | Index Definition |
| --- | --- | --- |
| **Health Care** | Health management | Providing health talks | To provide health knowledge lectures, to meet the community elderly health knowledge learning and daily care needs |
| | | Providing rehabilitation guidance | Provide rehabilitation nursing guidance to meet the needs of community elderly rehabilitation medication and daily care |
| | | Providing regular physical examination | Provide a regular physical examination service to ensure effective monitoring of the health status of elderly people in the community |
| | | Providing maintenance for chronic diseases | We will provide maintenance for chronic diseases, to ensure elderly people's safety and improve their quality of life |
| | Medical clinic | Providing an emergency rescue system | The emergency rescue system is set up to meet the needs of elderly people in emergency situations |
| | | Supporting community clinic | Community clinics were built to meet the daily medical needs of the elderly |
| | | Building community day care centers | Day care centers will be built in the community to provide sunshine and rehabilitation services for the elderly in need |
| | Family practice | Providing a dispensing and delivery service | Provide a dispensing and delivery service to solve the problem of getting medicine to the elderly |
| | | Providing on-site medical services | Provide a doctor's appointment to visit the doctor service, to solve the elderly's daily problems |
| | | Providing accompanying medical services | Provide family doctor services to ensure timely, effective and personalized medical care for the elderly |

**Table 4.** *Cont.*

| Coding Content | Evaluation Index | Index Definition |
|---|---|---|
| **Economic Security** | Life care | |
| | Building elderly canteen | Community supporting an "elderly canteen", to provide the community elderly with food and food delivery services |
| | Providing housekeeping services | The community provides housekeeping services for the elderly at home to improve the living environment of the elderly |
| | Providing life care services | Community to provide home care services for the elderly to improve the quality of life of the elderly |
| | Providing social security services | We will establish basic pension and medical insurance funds for elderly people in the community to ensure that sick elderly people receive necessary material help from society and reduce the burden of pension costs |
| | Consultation service | |
| | Public service information content | Community service projects, charging standards, and other basic information public, giving a comprehensive and true reflection of community services |
| | Providing policy and current affairs consultation | To provide a policy and current affairs consulting service, to give advice or solutions to problems in the community elderly people's life |
| | Providing service intermediary consultation | To provide consultation, assessment, and brokerage services for the daily trading activities of the community to ensure the rights and interests of the elderly in the community |
| | Internet pension | |
| | Establish elderly health information files | For the community elderly to establish detailed health and other aspects of life information files, to meet the community elderly health-management information resource needs |
| | Building a safety monitoring system | Through the real-time monitoring network system, the abnormal behaviors of the elderly are evaluated and identified to provide security for the elderly in the community |
| | Providing Internet retirement information | To provide a more comprehensive elder-care information and other information services for the community elderly to build a community elder-care at ease of the integrated service platform |
| | Providing an online clinic service | The online health management platform enables the community to communicate with doctors in real time and provide online diagnosis and treatment for patients |

*3.2. The Second Stage: The Selection of the Assessment Indicators for the Construction of Community Suitable for Aging under the Model of Community Home Care*

This stage continues the analysis of the previous text data. Through the collection and coding of the assessment indicators of the construction of communities suitable for aging, the collected data was used to develop the expert questionnaire. This study adopted the fuzzy Delphi expert questionnaire, including the first round of fuzzy Delphi expert questionnaire design and distribution, verification value, and expert consensus value calculation. The second round of the fuzzy Delphi expert questionnaire was issued while the convergence, the consensus value of experts, was calculated, and the indicators that did not reach the value threshold were deleted, so as to determine the hierarchical structure of communities suitable for aging construction evaluation index under the community home-care mode.

### 3.2.1. The Fuzzy Delphi Expert Questionnaire Design

In this study, the fuzzy Delphi method was applied to understand the "framework of indicators for the assessment of the construction of communities suitable for aging". Based on the previous text collection and coding, the initial "community suitable for aging construction evaluation indicators" were drawn up. In this stage, the "community suitable for aging construction evaluation indicators" and their hierarchical network architecture were established by the fuzzy Delphi expert questionnaire survey, and unnecessary indicators were deleted. In this study, experts' opinions were collected by means of a network questionnaire, that is, according to the indicators preliminarily drawn up, experts and scholars were asked to give subjective evaluation scores, so as to obtain the evaluation values of each indicator. The design of the questionnaire content is shown in Table 5.

**Table 5.** The fuzzy Delphi expert questionnaire description.

| Questionnaire Content | Survey Objective |
|---|---|
| 1. Filling explanation: Explain in detail how to fill in the questionnaire and illustrate with examples. | To make it easier for respondents to fill in the questionnaire and save time by using simple instructions. |
| 2. Description of evaluation indicators: It contains 4 primary codes, 14 secondary codes and 51 index factors. | To let the interviewees understand the structural relationship among the factors of the "community suitable for aging construction evaluation index system". |
| 3. Fill in the questionnaire and explain the indicators: The "optimal value" and "acceptable range" of the importance degree of each factor were filled in and scored by the experts one by one, and the evaluation indicators were briefly explained for the interviewees to understand the meaning of the indicators (such as in Table 6). | "Best value": please evaluate the importance of this indicator and fill in the best value for the importance of this indicator. "Acceptable range": please evaluate the acceptable range of importance of this indicator and fill in the acceptable maximum and minimum values. |
| 4. The last term, "other". | Open to experts and scholars to add indicators to supplement the initial list of indicators. |

**Table 6.** Examples of fuzzy Delphi expert questionnaires.

| Evaluate Index Items and Their Interpretation | Degree of Importance | Tolerance Interval | |
|---|---|---|---|
| | Optimum Value | Minimum Value | Maximum Value |
| **Evaluation indicators:** Participating in community management | | | |
| **Definition of indicators:** Encouraging the elderly to participate in the management of community residents' public affairs | 7 | 5 | 9 |

For example: according to an expert, the best value of the importance degree of the evaluation criteria of "participating in community management" is 7, the minimum value of the importance degree range is 5, and the maximum value is 9. The contents of the evaluation criteria are shown in Table 6.

### 3.2.2. Expert Selection and Questionnaire Distribution

In this study, it is believed that the assessment system of community fitness for aging under the model of community home care covers a wide and complex scope. If only a certain category of experts are used to comment on the issue of fitness for aging, people will doubt that the obtained data are reliable and unbiased. Therefore, it is necessary to include the opinions of diverse experts in

this study. According to Meltsner (1976), the technical indicators for selecting experts are "politics" and "analysis" [34]. In this study, experts and scholars with relevant research, teaching, and practical experience in environmental planning, social security, industrial design, industrial design, etc., will provide valuable suggestions during the investigation phase of this study, so as to achieve the appropriateness of the weight calculation of community fitness for aging evaluation. In the Delphi method, for the selection of the number of experts, Dalkey (1969) believed that the error margin of a group with a population of at least 10 people could be reduced to the minimum, with the highest reliability [35]. Therefore, the research committee invited 12 experts and scholars to participate in the questionnaire group. These experts consisted of eight males and four females, among whom three had a bachelor's degree, six had a master's degree and three had a doctor's degree. The interviewees included professional research experts at university professorship level, professional education experts at associate professorship level or above, and practical experts at director level or above in professional companies, who have been working for more than 20 years. The statistics of the expert data table are shown in Table 7.

**Table 7.** Expert Basic Statistics.

| Expert Member | Number of People | Proportion | Gender /Number | Education Level/Number | Profession/Number |
|---|---|---|---|---|---|
| Professional research | 4 | 33.3% | Male/3 Female/1 | Bachelor/3 Doctor/1 | Industrial design/2 Social Security/2 |
| Professional teaching | 4 | 33.3% | Male/2 Female/3 | Master/2 Doctor/2 | Environmental planning/2 Environmental design/1 Industrial design/1 |
| Professional practice | 4 | 33.3% | Male/3 Female/ 1 | Master/4 | Environmental planning/2 Architectural design/2 |

The Delphi questionnaire is conducted by asking experts to evaluate and score the importance of each indicator according to their personal subjective value. Generally speaking, the Delphi questionnaire usually only needs two rounds (Lin and Ren, 2009) [36]. This study also aims to conduct two rounds of the questionnaire survey, starting on 10 April 2020 and ending on 20 May 2020, respectively. A total of 24 questionnaires were issued twice, and 24 questionnaires were obtained with a valid rate of 100%.

3.2.3. Questionnaire Collection and Data Analysis

The fuzzy Delphi method is to introduce the fuzzy theory into the general Delphi method, integrate the expert opinions with the fuzzy triangular number method, and judge whether the expert opinions have reached convergence with the grey relational degree. The flatness of expert opinions can only be calculated after the convergence of opinions (Murray et al., 1985) [37]. The calculation steps are as follows:

Step 1: By setting interval values for the evaluation indicators to be considered, the positive cognitive value ($O^i$), and the conservative cognitive value ($C^i$) can be obtained. The higher the score is, the more important the index is.

Step 2: Statistical experts give a certain evaluation index conservative cognitive value and optimistic cognitive value, remove the extreme values that fall outside the "two standard deviations". Then calculate the minimum value ($C_L^i$), geometric average value ($C_M^i$), and maximum value ($C_U^i$) of conservative cognitive value in the remaining indicators respectively. The minimum ($O_L^i$), geometric mean ($O_M^i$), and maximum ($O_U^i$) of positive cognitive values are then calculated.

Step 3: The conservative cognitive triangular fuzzy number of each evaluation index was calculated $C^i = \left(C_L^i, C_M^i, C_U^i\right)$ as was the positive cognition triangle fuzzy number $O^i = \left(O_L^i, O_M^i, O_U^i\right)$

Step 4: Calculating the grey zone test value $M^i\left(O_M^i - C_M^i\right) - Z^i\left(C_U^i - O_M^i\right)$ to test whether experts agree on the evaluation indicators. When the result is positive $\left(M^i - Z^i \geq 0\right)$, it means that the expert

opinion tends to be consistent and the evaluation index reaches convergence, On the contrary, it means that the expert opinion of this indicator does not reach consensus, and the evaluation indicator needs to be carried out in the next round of expert investigation.

Step 5: When the experts agree the expert consensus value ($G^i$), the threshold value S is determined according to expert opinions or relevant standards for indicator screening.

This study was performed by Microsoft Excel 2010. A statistical analysis was carried out on 51 questions of the evaluation indexes of the construction of community suitable for aging under the model of community home care. The consensus value of experts was calculated, the indicators were screened, and the opinions of experts were revised.

## 4. Results

Through the analysis of policy texts, literature materials, and interview data, this study collected and coded the indicators of age-appropriate community construction under the model of community home-based care. In addition, two rounds of fuzzy Delphi expert questionnaires were used to screen the evaluation indicators of construction of communities suitable for aging. The research results and discussion are as follows.

*4.1. Calculation Results and Processing of Expert Questionnaires*

The first round of expert questionnaire survey was completed from 10 April 2020 to 1 May 2020, with 12 questionnaires issued and 12 returned. After the questionnaire was collected, indictors of statistical analysis of the data were carried out immediately. The calculation results showed that in the "improving neighborhood relations" and "providing an online clinic service" indicators, there is overlap between the two triangles, that is $C_U^i > O_L^i$, and grey areas of fuzzy relationships $Z^i = C_U^i - O_L^i > M^i = O_M^i - C_M^i$. It indicates that there is no consensus in the value of expert opinion interval, and that the difference between the expert who gives an extreme opinion value and other experts is too large, resulting in divergence of opinions and failure to reach consensus. Then the results of the first round of expert questionnaire analysis table were provided to all experts for reference, and the second round of the expert questionnaire survey was repeated. From 2 May 2020 to 20 May 2020, all the 12 questionnaires were returned. The analysis results of the second round of expert questionnaires are shown in Table 8.

After revision, the verification value ($M^i - Z^i$) of all evaluation items in the second round of the expert questionnaire is greater than 0, reaching a state of convergence, indicating that expert opinions reached consensus.

**Table 8.** Fuzzy Delphi expert questionnaire results (round 2).

| Evaluation Index | Conservative Values | | | Optimistic Values | | | $M^i - Z^i$ | $G^i$ |
|---|---|---|---|---|---|---|---|---|
| | $C_L^i$ | $C_M^i$ | $C_U^i$ | $O_L^i$ | $O_M^i$ | $O_U^i$ | | |
| Participating in community management | 3 | 4.64 | 7 | 7 | 8.10 | 10 | 3.46 | 6.37 |
| Organizing community activity | 4 | 5.16 | 7 | 7 | 8.62 | 10 | 3.46 | 6.89 |
| Exchanging community information | 3 | 5.20 | 8 | 8 | 8.39 | 10 | 3.19 | 6.80 |
| Improving community services | 4 | 5.70 | 8 | 7 | 9.12 | 10 | 2.42 | 7.41 |
| Providing spiritual comfort | 4 | 5.48 | 8 | 7 | 8.79 | 10 | 2.31 | 7.14 |
| Improving neighborhood relations | 3 | 5.01 | 8 | 6 | 8.12 | 10 | 1.11 | 6.57 |
| Providing psychological guidance | 3 | 5.20 | 8 | 7 | 8.77 | 10 | 2.57 | 6.99 |

**Table 8.** *Cont.*

| Evaluation Index | Conservative Values | | | Optimistic Values | | | $M^i - Z^i$ | $G^i$ |
|---|---|---|---|---|---|---|---|---|
| | $C_L^i$ | $C_M^i$ | $C_U^i$ | $O_L^i$ | $O_M^i$ | $O_U^i$ | | |
| Foster community identity | 4 | 5.15 | 8 | 7 | 8.35 | 10 | 2.20 | 6.75 |
| Providing cultural education | 3 | 4.82 | 7 | 7 | 8.36 | 10 | 3.54 | 6.59 |
| Providing skill training | 3 | 4.61 | 6 | 6 | 7.63 | 10 | 3.02 | 6.12 |
| Organizing cultural performances | 4 | 5.34 | 8 | 7 | 8.77 | 10 | 2.43 | 7.06 |
| Handling related affairs | 5 | 6.09 | 8 | 7 | 9.11 | 10 | 2.02 | 7.60 |
| Publicity policy | 4 | 4.92 | 7 | 6 | 8.18 | 10 | 2.26 | 6.55 |
| Managing supplies for the elderly | 3 | 5.08 | 7 | 7 | 8.28 | 10 | 3.20 | 6.68 |
| Providing legal aid | 4 | 5.40 | 7 | 7 | 8.62 | 10 | 3.22 | 7.01 |
| Improving fitness facilities | 4 | 5.29 | 8 | 7 | 8.79 | 10 | 2.50 | 7.04 |
| Providing leisure seating | 3 | 5.33 | 8 | 7 | 8.86 | 10 | 2.53 | 7.10 |
| Improving lighting facilities | 3 | 5.26 | 8 | 7 | 8.78 | 10 | 2.52 | 7.02 |
| Improving sanitation facilities | 3 | 4.99 | 8 | 6 | 8.82 | 10 | 1.83 | 6.91 |
| Road accessibility | 4 | 6.30 | 8 | 7 | 9.28 | 10 | 1.98 | 7.79 |
| Management of community vehicles | 4 | 5.74 | 8 | 7 | 8.93 | 10 | 2.19 | 7.34 |
| Improving community roads | 5 | 5.63 | 7 | 7 | 8.94 | 10 | 3.31 | 7.29 |
| Improving physical environment | 5 | 5.77 | 8 | 7 | 8.95 | 10 | 2.18 | 7.36 |
| Building facilities for the elderly | 3 | 5.87 | 8 | 7 | 9.28 | 10 | 2.41 | 7.58 |
| Space static separation | 3 | 5.07 | 8 | 7 | 8.44 | 10 | 2.37 | 6.76 |
| Formulate safety measures | 3 | 5.52 | 7 | 8 | 9.14 | 10 | 4.62 | 7.33 |
| Beautify community environment | 3 | 4.98 | 6 | 7 | 8.54 | 10 | 4.56 | 6.76 |
| Ensuring space accessibility | 4 | 5.31 | 7 | 7 | 8.70 | 10 | 3.39 | 7.01 |
| Supporting living facilities | 3 | 5.23 | 8 | 8 | 9.13 | 10 | 3.90 | 7.18 |
| Ensuring public transportation | 3 | 5.40 | 7 | 8 | 9.22 | 10 | 4.82 | 7.31 |
| Providing health talks | 3 | 5.21 | 7 | 8 | 8.97 | 10 | 4.76 | 7.09 |
| Providing rehabilitation guidance | 3 | 5.19 | 8 | 8 | 8.89 | 10 | 3.70 | 7.04 |
| Providing regular physical examination | 4 | 5.95 | 8 | 7 | 9.21 | 10 | 2.26 | 7.58 |
| Providing maintenance for chronic diseases | 4 | 5.89 | 8 | 9 | 9.32 | 10 | 4.43 | 7.61 |
| Providing emergency rescue system | 4 | 6.40 | 8 | 8 | 9.38 | 10 | 2.98 | 7.89 |
| Supporting community clinic | 3 | 5.91 | 8 | 7 | 9.29 | 10 | 2.38 | 7.60 |
| Building community day care centers | 4 | 5.26 | 7 | 7 | 8.78 | 10 | 3.52 | 7.02 |
| Providing a dispensing and delivery service | 3 | 5.62 | 7 | 8 | 9.14 | 10 | 4.52 | 7.38 |
| Providing on-site medical services | 4 | 5.74 | 8 | 8 | 9.31 | 10 | 3.57 | 7.53 |
| Providing accompanying medical services | 4 | 5.01 | 7 | 7 | 8.61 | 10 | 3.60 | 6.81 |

**Table 8.** *Cont.*

| Evaluation Index | Conservative Values | | | Optimistic Values | | | $M^i - Z^i$ | $G^i$ |
|---|---|---|---|---|---|---|---|---|
| | $C_L^i$ | $C_M^i$ | $C_U^i$ | $O_L^i$ | $O_M^i$ | $O_U^i$ | | |
| Building an elderly canteen | 3 | 5.20 | 8 | 7 | 8.78 | 10 | 2.58 | 6.99 |
| Providing housekeeping services | 3 | 5.06 | 7 | 7 | 8.61 | 10 | 3.55 | 6.84 |
| Providing life-care services | 3 | 4.97 | 7 | 7 | 8.60 | 10 | 3.63 | 6.79 |
| Providing social security services | 3 | 5.12 | 8 | 7 | 8.71 | 10 | 2.59 | 6.92 |
| Public service information content | 3 | 5.03 | 8 | 7 | 8.78 | 10 | 2.75 | 6.91 |
| Providing policy and current affairs consultation | 3 | 4.58 | 6 | 6 | 7.84 | 10 | 3.26 | 6.21 |
| Providing service intermediary consultation | 3 | 4.91 | 7 | 7 | 8.35 | 10 | 3.44 | 6.63 |
| Establish elderly information files | 4 | 5.42 | 8 | 8 | 8.64 | 10 | 3.22 | 7.03 |
| Building safety monitoring system | 3 | 5.30 | 7 | 7 | 8.59 | 10 | 3.29 | 6.95 |
| Providing Internet retirement information | 3 | 5.24 | 7 | 7 | 8.25 | 10 | 3.01 | 6.75 |
| Providing an online clinic service | 4 | 5.34 | 8 | 7 | 8.47 | 10 | 2.13 | 6.91 |

*4.2. Setting of Threshold Value and Construction of Evaluation Index System*

The size of the threshold value directly affects the selection of evaluation indicators, and the existing literature on how to determine the threshold value is based on the subjective judgment of researchers' experience. In consideration of the efficiency and cost of practical application, 80% or more than 90% of the index items approved by experts can be adopted (Chen, 2000) [38]. In order to make the results of the index screening more convincing and reasonable, this study applied the scatter diagram to analyze the steep slope and decided to set the threshold value of the expert consensus value to S ≥ 6.5 (Figure 1). The results showed that "participating in community management", "providing skill training " and "providing policy and current affairs consultation" did not reach the threshold value.

Based on the root coding of text data and two rounds of expert questionnaires of the fuzzy Delphi method, the indexes that did not reach the threshold value were deleted. Finally, the study determined the framework of the "assessment index system for the construction of community fitness for aging under the model of community home care", as shown in Figure 2.

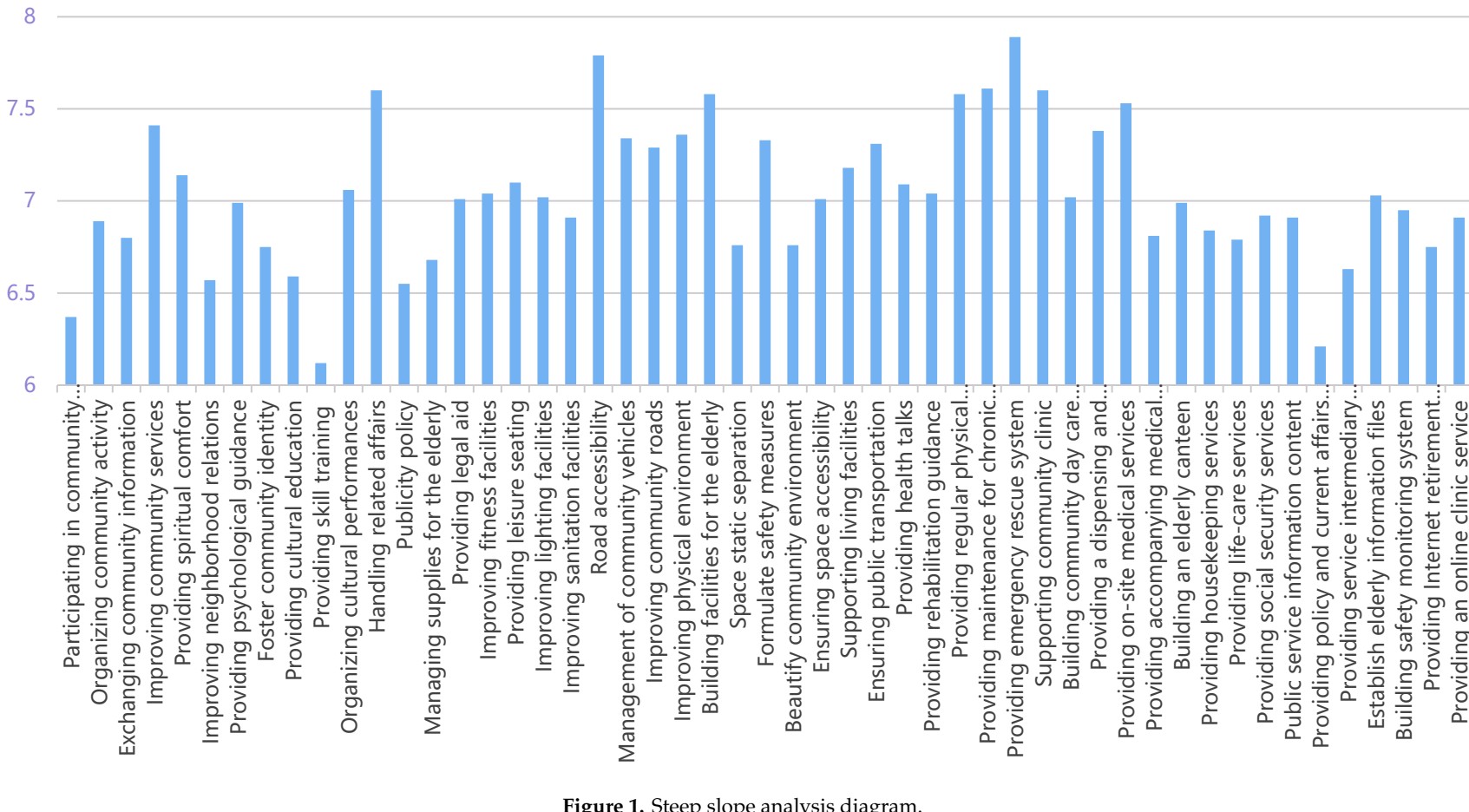

**Figure 1.** Steep slope analysis diagram.

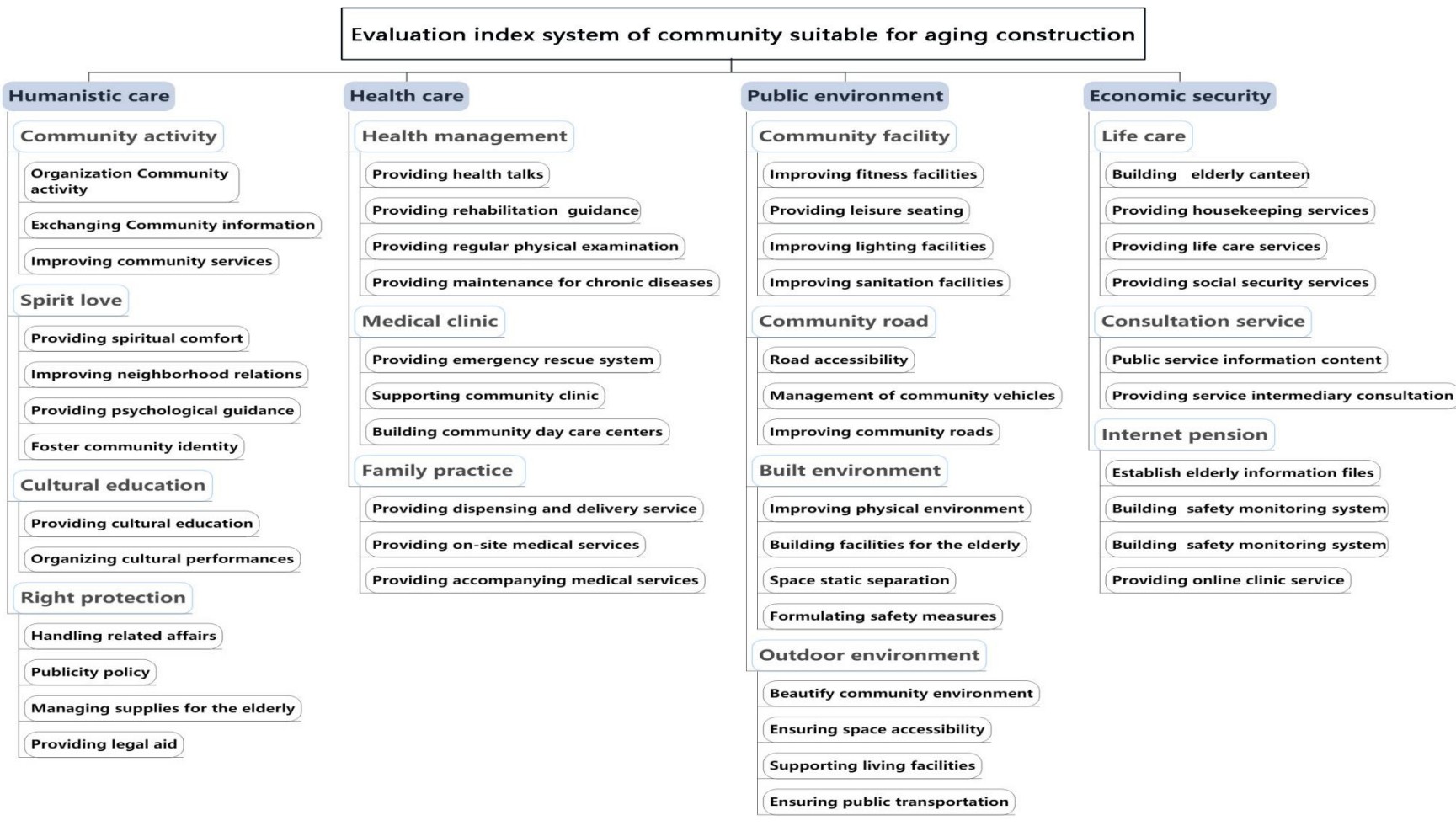

**Figure 2.** Evaluation index system of construction of communities suitable for aging.

### 4.3. Results of Expert Consensus Values

Through the evaluation of the expert questionnaire, in the four-criteria layer, the consensus value of experts was ranked as "health care (7.35)" > "public environment (7.18)" > "humanistic care (6.92)" > "economic security (6.85)". Among the "humanistic care" criteria, "community activities (7.03)" scored the highest. "Community roads (7.47)" scored highest on the "public environment" criteria level. In the "health care" criteria layer, the "medical treatment (7.50)" score is the highest. Among the "economic security" criteria, "Internet pension (6.91)" scored the highest. Of the 48 overall evaluation indicators, the top three that deserve the highest marks are "providing an emergency rescue system (7.89)", "road accessibility (7.79)" and "handling related affairs (7.60)". The calculation results of the consensus value of the indicators for the assessment of community fitness for aging are shown in Table 9.

**Table 9.** Expert consensus value calculation table.

| Criterion Layer | Sub-Criterion Layer | Evaluation Index | Expert Consensus Value | | | |
|---|---|---|---|---|---|---|
| Humanistic care | Community activity | Organizing community activity | 6.89 | | | |
| | | Exchange community information | 6.80 | 7.03 | | |
| | | Improving community services | 7.41 | | | |
| | Spiritual care | Providing spiritual comfort | 7.13 | | | |
| | | Improving neighborhood relations | 6.56 | 6.85 | | |
| | | Providing psychological guidance | 6.98 | | | |
| | | Foster community identity | 6.75 | | 6.92 | |
| | Cultural education | Providing cultural education | 6.59 | 6.82 | | |
| | | Organizing cultural performances | 7.05 | | | |
| | Right protection | Handling related affairs | 7.60 | | | |
| | | Publicity policy | 6.55 | 6.95 | | |
| | | Managing supplies for the elderly | 6.68 | | | |
| | | Providing legal aid | 7.01 | | | |
| Public environment | Community facility | Improving fitness facilities | 7.04 | | | 7.07 |
| | | Providing leisure seating | 7.10 | 7.02 | | |
| | | Improving lighting facilities | 7.02 | | | |
| | | Improving sanitation facilities | 6.90 | | | |
| | Community road | Road accessibility | 7.79 | | | |
| | | Management of community vehicles | 7.33 | 7.47 | | |
| | | Improving community roads | 7.29 | | 7.18 | |
| | Built environment | Improving physical environment | 7.36 | | | |
| | | Building facilities for the elderly | 7.57 | 7.25 | | |
| | | Space static separation | 6.76 | | | |
| | | Formulate safety measures | 7.33 | | | |
| | Outdoor environment | Beautify community environment | 6.76 | | | |
| | | Ensure space accessibility | 7.01 | 7.06 | | |
| | | Supporting living facilities | 7.18 | | | |
| | | Ensure public transportation | 7.31 | | | |

**Table 9.** *Cont.*

| Criterion Layer | Sub-Criterion Layer | Evaluation Index | Expert Consensus Value | | |
|---|---|---|---|---|---|
| Health care | Health management | Providing health talks | 7.09 | | |
| | | Providing rehabilitation guidance | 7.04 | 7.32 | |
| | | Providing regular physical examination | 7.58 | | |
| | | Providing maintenance for chronic diseases | 7.60 | | |
| | Medical clinic | Providing an emergency rescue system | 7.89 | | 7.35 |
| | | Supporting community clinic | 7.60 | 7.50 | |
| | | Building community day care centers | 7.02 | | |
| | Family practice | Providing a dispensing and delivery service | 7.38 | | |
| | | Providing on-site medical services | 7.52 | 7.23 | |
| | | Providing accompanying medical services | 6.81 | | |
| Social economy | Life care | Building an elderly canteen | 6.99 | | |
| | | Providing housekeeping services | 6.84 | 6.88 | |
| | | Providing life-care services | 6.79 | | |
| | | Providing social security services | 6.91 | | |
| | Consultation service | Public service information content | 6.90 | | 6.87 |
| | | Providing service intermediary consultation | 6.63 | 6.76 | |
| | Internet pension | Establish elderly health information files | 7.03 | | |
| | | Building a safety monitoring system | 6.95 | 6.91 | |
| | | Provide Internet retirement information | 6.74 | | |
| | | Provide an online clinic service | 6.90 | | |

## 5. Discussion

### 5.1. Discussion on the Consensus Value of Experts at the Criterion Level

According to the calculation of the consensus value of the fuzzy Delphi method, the results show that the important values in the four criteria are "health care" (7.35), "public environment" (7.18), "humanistic care" (6.92), and "economic security" (6.87). Studies have shown that with the aging of elderly people in community home care and the decline of physical function, the demand for healthcare security is increasing. However, according to the current investigation, there is a lack of correlation between the community living of the elderly and their community medical institutions, How to vigorously develop the community medical system with the community as the carrier, establish health and medical security conditions for the elderly living at home in the community, and ensure the physical and mental health of the elderly and the quality of life of the sick elderly, it is an important material basis for the development of community home-based care model (Dai and Zhou, 2019) [39].

### 5.2. Discussion on the Consensus Value of Experts at the Sub-Criterion Level

In the criterion layer of "health care", the sub-criterion of "medical treatment (7.50)" has the highest score. The community home-care model has become the mainstream of China's old-age care model, but there is a lack of medical care for the elderly. The elderly have a strong demand for local medical treatment, and it is an urgent social problem to build an appropriate system combining medical care with nursing care (Dai and Zhou, 2019) [39]. In the "public environment" criteria layer, the "community roads (7.47)" sub-criterion scored the highest. Due to the limited activity capacity and limited activity range of the elderly, the community road space is an important place for the elderly to strengthen their bodies and have a rest from walking. The construction of community public environments suitable for aging should focus on the construction of outdoor community road space

for the elderly with community parks, green spaces, and squares as the core, so as to improve the living environment of the elderly and form a structure of living space suitable for the elderly (Xie, Wei, and Zhou, 2015) [40]. In the criteria layer of "humanistic care", the sub-criterion of "Community activities (7.03)" has the highest score. To accelerate the development of community home-care service, community activities must be organized to strengthen community influence, so that the elderly can better understand the community and get familiar with community services, and strengthen their sense of identity to community home care (Gao, 2011) [41]. In the criterion layer of "economic security", "Internet pension (6.91)" has the highest score. Today, with the increasingly far-reaching influence of the Internet on all walks of life, the introduction of the Internet mode in the community home care industry is a necessary measure to alleviate the imperfect construction of communities suitable for aging and the imbalance in the supply of elder-care services (Sui and Peng, 2016) [42].

*5.3. Discussion on the Consensus Value of Experts at the Evaluation Index*

Of the 48 community building assessment indicators, "setting up an emergency assistance system" (7.89), "ensuring barrier-free roads" (7.79) and "handling related matters" (7.60) ranked the highest. This shows that the protection of the physical and mental health of the elderly and the protection of the social rights and interests of the elderly are the core content of the construction of the community suitable for the aging. The elderly are a group with fragile health. Due to chronic diseases, self-care disorders, cognitive decline, and psychological changes affecting their health and lives, their ability to control the environment declines, and their ability to deal with environmental emergencies also declines, and they are prone to many safety problems. "Setting up an emergency assistance system" and "ensuring barrier-free roads" are important measures to ensure the daily security of the elderly in the community (Lu, 2012) [43].

## 6. Conclusions

This study used the fuzzy Delphi expert questionnaire survey method to select the assessment indicators of communities suitable for aging construction, so as to complete the construction of the communities suitable for aging evaluation index system. Competent government departments and community home-care workers can use assessment tools to effectively screen the construction indicators suitable for aging and make decisions, so as to find the suitable aging performance that meets the needs of elderly people living at home in the community. The conclusions of this study are described as follows.

*6.1. Research Conclusions*

In this study, based on the text data based on the code and the fuzzy Delphi expert questionnaire survey, a hierarchical structure of indicators for the assessment of community fitness for aging construction was obtained, including 4 criterion levels of humanistic care, public environment, health care and economic security; 14 sub-criteria; and a total of 48 detailed indicators. These criteria and indicators are the key factors affecting the assessment of community fitness for aging. Through these detailed indicators, the construction of community fitness for aging under the model of community home care can be evaluated more completely and objectively.

In this study, expert consensus values of 4 criteria, 14 sub-criteria, and 48 detailed indicators were obtained by fuzzy Delphi questionnaires. The results show that the two criteria of "health care" and "public environment" are the core criteria for evaluating the conditions of community aging construction, as long as the community aging construction is involved in the new community or the old community reconstruction (Zhao, 2019) [44]. The guidelines of "humanistic care" and "community economy" are more of an effective supplement to the community aging construction.

In the past, researches on the community home care model focused on the existing problems and suggestions in the construction and management of the elder-care service system and the community living environment, while few data related to the evaluation index system of communities suitable

for aging. Through the national government standard, the relevant research literature, the old root of interview data, combined with fuzzy Delphi expert questionnaire, and rigorous research, this study has the objectivity of the evaluation index and the important values of every index for production in the future academic community-aging rules of construction in the planning stage, to provide a reference.

*6.2. Research Suggestion*

This study has preliminarily established the evaluation index system of community fitness for aging construction, and the future research direction will take this as the basis for the verification analysis of actual cases, in order to test whether the evaluation index system of community fitness for aging construction constructed by this research can effectively evaluate the effect of a single community or multiple communities' fitness for aging construction.

This research mainly adopts domestic standards and expert opinion, and suggests that the standards of developed countries and different nationalities assess the optimum aging community construction of expert opinion, not only to broaden the community aging evaluation index research more comprehensively and contribute to the development of community home endowment of globalization strategy, but also further to analyze the differences of domestic and foreign experts views on this issue.

This study mainly focuses on the evaluation index system of aging suitability construction of urban communities under the home-based care model. It is suggested that the scope of the study can be extended to the conditions of aging suitability in the construction of communities in rural areas. It can not only improve the evaluation index system of community aging suitability construction, but also contribute to the development of comprehensive strategies of community home-based care, and further analyze the differences of elderly people's requirements for community aging construction in urban and rural areas.

This study attempts to construct a comprehensive evaluation system of community suitable for aging. The construction of any evaluation system will inevitably encounter the dilemma of universality and particularity. This study tries to take into account the general and local needs. Through the national policy text, professional literature, the elderly interview records, and other community aging construction needs of the data collection, as well as the questionnaire survey of experts in the fields of community management, social security and environmental construction, it is hoped to construct a comprehensive construction evaluation system that can reflect the operation condition of community aging. However, it should be noted that the establishment of the community aging suitability assessment system should be a dynamic interactive process, which needs to be continuously modified by the decision makers and the elderly in the community. Due to the limitations of research resources, number of experts invited, and research methods, this study is only a preliminary exploration, and more detailed investigation and analysis are needed to modify the feedback of the evaluation system in the future.

**Author Contributions:** Conceptualization, W.-B.M. and S.-J.O.; methodology, W.-B.M. and S.-J.O.; software, W.-B.M.; S.-J.O. and C.-Y.H.; validation, S.-J.O., W.-B.M. and C.-Y.H.; formal analysis, S.-J.O., W.-B.M. and C.-Y.H.; investigation, W.-B.M. and S.-J.O.; data curation, W.-B.M. and C.-Y.H.; writing—original draft preparation, W.-B.M. and S.-J.O.; writing—review and editing, W.-B.M.; S.-J.O. and C.-Y.H.; supervision, S.-J.O.; project administration, W.-B.M.; S.-J.O. and C.-Y.H. All authors have read and agreed to the published version of the manuscript.

**Funding:** This research received no external funding

**Acknowledgments:** 

**Conflicts of Interest:** The authors declare no conflict of interest

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
