# Peer review of "Research on the Evaluation Index System of the Construction of Communities Suitable for Aging by the Fuzzy Delphi Method"

_environments, doi:10.3390/environments7100092_

Round 1
Reviewer 1 Report
This manuscript discussed the issue of evaluation index system of community suitable for aging. This issue is essential for aging in place.
This paper is difficult to follow because of these main reasons:
- Lacking sufficient literatures for theoretical discussion.
- Definition of criterion layers should be considered carefully.
There are some comments that allow improving the work:
- In section of literature review, authors’ references almost focused in researches from China. Also, references to previous recent research are marginal. For global readers, the comprehensive evaluation of literature review is very important. Therefore, references can’t provide strong supports for constructing evaluation index system of community suitable for aging.
- My second concern is the first stage in “3.1.1. Policy text analysis”. Why are national policies texts selected to be the first stage work? Why do authors think official documents could depict elderly’s consideration of suitable community? Therefore, I don’t understand why Table 1 could induce these four criterion layers.
- The third comment is about the definition of “social economy”. In this paper, authors defined the sub-criterion layers of “social economy” were composed of “life care”, “consultation service”, and “internet pension”. I suggest authors should carefully use the term of “social economy”. The social economy is generally taken to be a third sector of mixed capitalist economies distinct from the private and public sectors. The social economy is based on cooperative, non-profit, and voluntary rather than paid activities carried out within communities, across national economies, and internationally. Many social economy organizations simply deliver services to their members or others they aim to serve without making use of the market.
Author Response
Point 1:In section of literature review, authors’ references almost focused in researches from China. Also, references to previous recent research are marginal. For global readers, the comprehensive evaluation of literature review is very important. Therefore, references can’t provide strong supports for constructing evaluation index system of community suitable for aging.
Response 1: According to the experts' opinions, the literature review has added the latest research results of different evaluation systems and research methods of old-age care indicators, such as Active Aging Index(AAI) and Selfie Aging Index(SAI).
Please see line148-190 of the revised paper
Point 2: My second concern is the first stage in “3.1.1. Policy text analysis”. Why are national policies texts selected to be the first stage work? Why do authors think official documents could depict elderly’s consideration of suitable community? Therefore, I don’t understand why Table 1 could induce these four criterion layers.
Response 2: Community home-based care for the aged is a policy formulated and implemented by the Chinese government according to the situation of the aging society, which is the background of this paper. How to help the elderly in the community to age successfully is not only a problem facing the family and the community, but also a responsibility of the whole society. In the past decades, Chinese government departments have successively issued relevant policy texts to guide the aging construction of communities. Therefore, this paper chooses the study of national policy texts as the first stage of work. In this paper, the content of 4 standard layers is obtained through comprehensive coding and literature research.
Please see line244-254 of the revised paper
Point 3: The third comment is about the definition of “social economy”. In this paper, authors defined the sub-criterion layers of “social economy” were composed of “life care”, “consultation service”, and “internet pension”. I suggest authors should carefully use the term of “social economy”. The social economy is generally taken to be a third sector of mixed capitalist economies distinct from the private and public sectors. The social economy is based on cooperative, non-profit, and voluntary rather than paid activities carried out within communities, across national economies, and internationally. Many social economy organizations simply deliver services to their members or others they aim to serve without making use of the market.
Response 3: Thank you for your advice. “social economy” changes to “Economic Security” , Because it is concerned with endowment expenses

Reviewer 2 Report
The paper presents an interesting research which fits to the scope of the journal. The manuscript focus on important topic which is worth investigation. However, the paper should be improved before publishing. Below I present some comments.
- Considering the issue of “evaluation index” the literature review is missing the connection to existing evaluations at different level, like Active Ageing Index, Selfie Aging Index, etc. It is important to present existing approaches, highlight gaps, and based on that propose a new evaluation method in order to explain the reader how it enrich current systems.
- In the literature review there are elements of needs of older adults, however, I there is a lack of “urban” aspects, which is important from the point of view of this study. I suggest to enrich this section by strong reference to “urban ageing” (see for example: Urban ageing [in:] Indoor and Built Environment, 2018, Vol. 27(5) 583–586, and other works in that domain).
- Line 178: “soded”, please verify.
- Figure 1: please change a style of the chart. Line chart assume that there is a continuity of data (e.g. following years), and in this research individual factors were assessed. Bar chart seems to be more suitable for this purpose for instance. Additionally please adjust the size of the chart so that whole names of factors can be clearly seen.
- Figure 2: in order to increase the visibility of the figure, please check if placing it in horizontal size (at the whole page) would be helpful.
- There is a double citation scheme. Please adjust it according to the MDPI style.
- The manuscript is based on the very limited number of references. However, after considering 1st and 2nd comment it could be already enriched.
I encourage the Authors to correct the paper, as in my opinion it presents an interesting study and might constitute a valuable paper after improvements mentioned above.
Author Response
Response to Reviewer 2 Comments
Point 1:Considering the issue of “evaluation index” the literature review is missing the connection to existing evaluations at different level, like Active Ageing Index, Selfie Aging Index, etc. It is important to present existing approaches, highlight gaps, and based on that propose a new evaluation method in order to explain the reader how it enrich current systems.
Response 1: Thank you for your advice. According to the experts' opinions, the literature review has added the latest research results of different evaluation systems and research methods of old-age care indicators, such as Active Aging Index(AAI) and Selfie Aging Index(SAI).
Please see line148-190 of the revised paper
Point 2: In the literature review there are elements of needs of older adults, however, I there is a lack of “urban” aspects, which is important from the point of view of this study. I suggest to enrich this section by strong reference to “urban ageing” (see for example: Urban ageing [in:] Indoor and Built Environment, 2018, Vol. 27(5) 583–586, and other works in that domain).
Response 2: Thank you for your advice. I referred to the literature you recommended. In this paper, the research object is aimed at the needs of the elderly in urban "community home-based care". Therefore, most of the literature reviews are focused on the needs of the elderly in urban communities.
Point 3: Line 178: “soded”, please verify.
Response 3: It is should be “coded” .
Point 4: Figure 1: please change a style of the chart. Line chart assume that there is a continuity of data (e.g. following years), and in this research individual factors were assessed. Bar chart seems to be more suitable for this purpose for instance. Additionally please adjust the size of the chart so that whole names of factors can be clearly seen.
Response 4: Thank you for your comments. I adjusted the line graph to the bar graph and adjusted the size of the graph。
Point 5: Figure 2: in order to increase the visibility of the figure, please check if placing it in horizontal size (at the whole page) would be helpful.
Response5: It has been revised according to your opinions. Please refer to the revised draft for details.
Point 6: There is a double citation scheme. Please adjust it according to the MDPI style.
Response6:I have perfected the bibliography according to MDPI format.
Point 7:The manuscript is based on the very limited number of references. However, after considering 1st and 2nd comment it could be already enriched.
Response7:Yes, some references have been added according to the experts.

Reviewer 3 Report
Dear authors, first of all congratulations for this important study.
The manuscript, however, to have the quality that the quality of the study deserves, should have some improvements:
Abstract:
Revise english: For e.g. line 17 "... consists of 4 criteria ceg humanistic care..."
or line 27: "...through constructing the ecaluation index system of the community suitable for aging..
Introduction:
Revise the expression "aging society", maybe you want to mean "aged society".
Revise bibliographic references along the text. They are not as the journal rules. E.g.: (Tong, X 2015) [1]- I must be just [1] after the text.
I understand the option of dividing the text in different points, but maybe some of them are not necessary to separate... Just a suggestion.
In the end of introduction you should present the categories that your study will analyse in the index.
Material and methods
To make the policy texts and literature revision analyses, you made a categorization apriori? Is is not clear in the explaination so the reader can understand the tables presented. Can you please explaine it better in the text?
The Delphi panel is well explained but there is a need to explaine the critheria of inclusion on the study. Why that 12 participants?
Results and conclusions:
You should refere to the study limitations. The 12 participants in the delphi allow that the results can be expanded to other contexts?
Just a litle more effort and the manuscript will be much better.
I suggest you to ask for help in the translation. There is several english edition to be done.
Author Response
Abstract:Revise english: For e.g. line 17 "... consists of 4 criteria ceg humanistic care..."or line 27: "...through constructing the ecaluation index system of the community suitable for aging..
Response:Thank you for your review. I have revised this part and will improve my English writing.
Introduction:
Point 1: Revise the expression "aging society", maybe you want to mean "aged society".
Response 1:Yes, I want to mean “aging society”.
Point 2: Revise bibliographic references along the text. They are not as the journal rules. E.g.: (Tong, X 2015) [1]- I must be just [1] after the text.
Response 2:Yes, I have perfected the bibliography according to APA format.
Point 3: I understand the option of dividing the text in different points, but maybe some of them are not necessary to separate... Just a suggestion.
Response 3:Thank you for your review. It has been adjusted according to your opinion. Please see Line 74-81 of the revised paper
Point 4: In the end of introduction you should present the categories that your study will analyse in the index.
Response 4:Thank you for your review. It has been adjusted according to your opinion. Please see Line 74-81 of the revised paper
Material and methods
Point 1:To make the policy texts and literature revision analyses, you made a categorization apriori? Is is not clear in the explaination so the reader can understand the tables presented. Can you please explaine it better in the text?
Response 1:The source and selection method of free nodes are analyzed by supplementing the coding content of policy text and literature. Please see Line 245-254 of the revised paper
Point 2: The Delphi panel is well explained but there is a need to explaine the critheria of inclusion on the study. Why that 12 participants?
Response 2: The selection criteria of 12 participants were based on references 33 and 34. Please see Line 338-349 of the revised paper
Results and conclusions:
Point 1: You should refere to the study limitations. The 12 participants in the delphi allow that the results can be expanded to other contexts?
Response 1:In this paper, 12 experts only scored the importance degree of 51 suitable aging evaluation indexes. As a method of data research, Delphi method has its limitations just like other research methods
Point 2:Just a litle more effort and the manuscript will be much better. I suggest you to ask for help in the translation. There is several english edition to be done.
Response 2: I will work hard to improve my English writing and will ask the MDPI professional translators to help me with the corrections to reach the publishing level before the final publication

Round 2
Reviewer 1 Report
Some English language are minor spell check required.
Please see lines 246-251. For example, this sentence didn't have a verb and was too long for reading comprehension.
Author Response
Point 1:Some English language are minor spell check required.
Please see lines 246-251. For example, this sentence didn't have a verb and was too long for reading comprehension.
Response 1: I will work hard to improve my English writing and will ask the MDPI professional translators to help me with the corrections to reach the publishing level before the final publication

Reviewer 2 Report
I noticed significant improvements in the revised version of the paper. However, despite responses I still see that some elements were not corrected.
Ad. 2. In the response it is written “I referred to the literature you recommended.”, however, I do not see it in the manuscript.
Ad. 4. Previous recommendation includes “adjust the size of the chart so that whole names of factors can be clearly seen” but after correction still not all factors are named (half of them do not have a label and some ends with “…”).
Ad. 5. New version of the figure is much better, however, some boxes are smaller than text inside. Please correct it.
Author Response
Point 1: In the response it is written “I referred to the literature you recommended.”, however, I do not see it in the manuscript.
Response 1: Thank you for your advice,I referred to the literature you recommended.(The literature of 20)
The WHO project proposed that an age-friendly city is one that promotes active ageing and optimises opportunities for health, participation and security, in order to enhance quality of life as people age. The features of age-friendly cities were determined in eight domains of urban life, namely outdoor spaces and buildings; transportation; housing; social participation; respect and social inclusion; civic participation and employment; communication and information and community support and health services.
Point 2:Previous recommendation includes “adjust the size of the chart so that whole names of factors can be clearly seen” but after correction still not all factors are named (half of them do not have a label and some ends with “…”).
Response 2: Thank you for your advice, I'll adjust the format (Figure 1).
Point 3: New version of the figure is much better, however, some boxes are smaller than text inside. Please correct it.
Response 3: Thank you for your advice, I'll adjust the format (Figure 2).

Reviewer 3 Report
Dear authors. Good effort.
The manuscript is much better now.
It would be good to present the study limitations of the study in the results, as the study contributes to science and for society. This makes the results much unterstandable to the readers. Just a suggestion to be a five stars manuscript.
Author Response
Point 1:Dear authors. Good effort. The manuscript is much better now. It would be good to present the study limitations of the study in the results, as the study contributes to science and for society. This makes the results much unterstandable to the readers. Just a suggestion to be a five stars manuscript.
Response 1: Thank you for your advice,
This study attempts to construct a comprehensive evaluation system of community suitable for aging. The construction of any evaluation system will inevitably encounter the dilemma of universality and particularity, This study tries to take into account the general and local needs, Through the national policy text, professional literature, the elderly interview records and other community aging construction needs of the data collection, As well as the questionnaire survey of experts in the fields of community management, social security and environmental construction, it is hoped to construct a comprehensive construction evaluation system that can reflect the operation condition of community aging. However, it should be noted that the establishment of the community aging suitability assessment system should be a dynamic interactive process, which needs to be continuously modified by the decision makers and the elderly in the community. Due to the limitations of research resources, number of experts invited and research methods, this study is only a preliminary exploration, and more detailed investigation and analysis are needed to modify the feedback of the evaluation system in the future.

Round 3
Reviewer 2 Report
The paper has been corrected according to my previous comments and in my opinion it can be published in the current form.